# Troponin T Mutation as a Cause of Left Ventricular Systolic Dysfunction in a Young Patient with Previous Surgical Correction of Aortic Coarctation

**DOI:** 10.3390/biom11050696

**Published:** 2021-05-06

**Authors:** Martina Caiazza, Michele Lioncino, Emanuele Monda, Francesco Di Fraia, Federica Verrillo, Roberta Pacileo, Federica Amodio, Marta Rubino, Annapaola Cirillo, Adelaide Fusco, Emanuele Romeo, Alessandra Scatteia, Santo Dellegrottaglie, Paolo Calabrò, Berardo Sarubbi, Anwar Baban, Giulia Frisso, Maria Giovanna Russo, Giuseppe Limongelli

**Affiliations:** 1Inherited and Rare Cardiovascular Diseases, Department of Translational Medical Sciences, University of Campania “Luigi Vanvitelli”, Monaldi Hospital, 80131 Naples, Italy; martina.caiazza@yahoo.it (M.C.); michelelioncino@icloud.com (M.L.); emanuelemonda@me.com (E.M.); fyrent@libero.it (F.D.F.); fedeverrillo@gmail.com (F.V.); pacileoroberta@gmail.com (R.P.); amodio.federica@yahoo.it (F.A.); rubinomarta@libero.it (M.R.); cirilloannapaola@gmail.com (A.C.); adelaidefusco@hotmail.it (A.F.); mariagiovanna.russo@unicampania.it (M.G.R.); 2Adult Congenital Heart Disease Unit, Department of Cardiology, Monaldi Hospital, 80131 Naples, Italy; ema.romeo@alice.it (E.R.); berardo.sarubbi@virgilio.it (B.S.); 3Division of Cardiology “Villa dei Fiori” Hospital, Acerra, 80011 Naples, Italy; a.scatteia@gmail.com (A.S.); sandel74@hotmail.com (S.D.); 4Division of Clinical Cardiology, A.O.R.N. “Sant’Anna e San Sebastiano”, Department of Translational Medical Sciences, University of Campania “Luigi Vanvitelli”, 80131 Napoli, Italy; paolo.calabro@unicampania.it; 5Department of Pediatric Cardiology and Cardiac Surgery, Bambino Gesù Children’s Hospital and Research Institute, 00165 Rome, Italy; anwar.baban@opbg.net; 6Department of Molecular Medicine and Medical Biotechnology, Federico II University of Naples, CEINGE Scarl-Advanced Biotechnologies, 80131 Naples, Italy; giulia.frisso@unina.it; 7Institute of Cardiovascular Sciences, University College of London and St. Bartholomew’s Hospital, London WC1E 6DD, UK

**Keywords:** troponin T, aortic coarctation, left ventricular systolic dysfunction

## Abstract

Coarctation of the aorta is a leading cause of morbidity and mortality among adults with congenital heart disease (ACHD). Lifelong surveillance is mandatory to screen for possible long-term cardiovascular events. Left ventricular systolic dysfunction has been reported in association with recoarctation, and association with dilated cardiomyopathy (DCMP) is very rare. Herein, we report the case of a 19-year-old boy with coarctation of the aorta who complained of mild exertional dyspnea. Cardiac magnetic resonance revealed a moderately dilated, hypokinetic left ventricle (LV), with mildly reduced EF (45%), and residual isthmic coarctation was excluded. Genetic tests revealed a heterozygous missense variant in TNNT2 (NM_001001430.2): c.518G>A (p. Arg173Gln). This case highlights the role of careful history taking: a family history of cardiomyopathy should not be overlooked even when the clinical setting seems to suggest a predisposition to hemodynamic factors for LVSD.

## 1. Introduction

With an estimated prevalence of 1 in 2500 live births, coarctation of the aorta (COA) is a major cause of morbidity and mortality among adults with congenital heart disease (ACHD) [1]. Lifelong surveillance after surgical correction is mandatory, as increased risk of cardiovascular events is well documented, particularly after the third decade of life, including systemic hypertension, aortic valve abnormalities, and risk of aortic aneurysm [2]. Although it has been widely accepted that increased left ventricle (LV) afterload plays a major role in cardiovascular events, to date, there is a lack of data about the association of COA and left ventricular systolic dysfunction (LVSD) [3,4]. Few cases of LVSD have been reported in literature and mostly, recoarctation has been considered causative, as hemodynamic relief is associated to complete regression of LVSD [5].

Herein we report the case of a 19-year-old boy who was referred to our clinic because of unexplained LVSD and positive family history of dilated cardiomyopathy (DCM).

## 2. Case Report

The proband is the third son of non-consanguineous parents. At the age of 19, he was referred to our clinic for a comprehensive evaluation after he experienced mild exertional dyspnea in the last few weeks. The family history was positive to sudden death (his cousin died, aged 3 months) and to DCMP and heart transplant (HT): a maternal aunt and her son who then died at 42 and 24 years old, respectively. Figure 1 summarizes the family pedigree. At birth, he was diagnosed with COA and therefore surgically corrected by end-to-end anastomosis and patch aortoplasty. Since then, he had been routinely evaluated at the Pediatric Cardiology Unit in our Hospital clinic with serial clinical and echocardiographic evaluations, which showed normal myocardial function until our evaluation. He had normal growth parameters (weight 85 Kg and height 165 cm) and reached regularly developmental milestones suggesting an isolated form of COA.

During routine evaluation, the patient complained of progressive exertional dyspnea, NYHA II-III. There was no medical history for recent infectious nor inflammatory events. On physical examination: lower limb pulses were bilaterally palpable, there was no significant upper and lower limb pressure differences and there were no signs of peripheral congestion (neither crepitation nor hepatomegaly). Echocardiographic examination showed a mildly reduced left ventricular ejection fraction (LVEF) (∼42%) with nondilated LV. Residual isthmic gradient was not significant (<20 mmHg) and absence of holodiastolic run off was ascertained. Echocardiography excluded other findings that may suggest hemodynamic consequences. The aortic valve was tricuspid and no mitral valve abnormalities were found. Ambulatory blood pressure monitoring excluded hypertensive status suggesting recoarctation. Complete blood count, thyroid hormones, serum electrolytes, lipid profile, hepatic function, serum creatinine and NT pro BNP laboratory values were within the reference ranges. Medical therapy was introduced and well tolerated with beta-blocker (Bisoprolol) and ACE inhibitor (Ramipril) [6]. The patient was referred to the Inherited and Rare Diseases Unit on the suspicion of an underlying cardiomyopathy. For tissue characterization, a cardiac magnetic resonance (CMR) was performed showing a moderately dilated, hypokinetic left ventricle (LV), with mildly reduced EF (45%). T2-weighted images and double inversion recovery sequences excluded myocardial edema. Contrast enhanced fast spin echo T1-weighted sequences did not show any abnormality. No evidence of early and late gadolinium enhancement (LGE) was detected and residual isthmic coarctation was excluded (Figure 2).

## 3. Results

Based on the clinical presentation and family history, genetic counselling was performed and molecular testing was suggested. According to the dispositions of the local ethics committee, informed consent was appropriately obtained for genetic investigations through next generation sequencing (NGS) with a panel of 325 genes, known to be associated with cardiomyopathies (Appendix A).

A blood sample in EDTA was collected from the subject. Genomic DNA was extracted using the Maxwell 16 instrument (Promega, Madison, WI, USA), and DNA quality was assessed using a Nanodrop machine. Molecular testing was carried out by analyzing a panel of target genes through NGS-based procedure. The Alyssa software (Agilent, Santa Clara, CA, USA) was used to perform sequence data analysis. This tool allows the alignment of the sequences to the reference genome in order to obtain a list of genomic variants that can be prioritized using a bioinformatic pipeline in order to highlight likely pathogenic or pathogenic variants. Genome data processing was performed using a home-made bioinformatic pipeline. The MAF threshold was set to 5% using the Illumina Variant Interpreter Software. Genetic testing identified a heterozygous variant in TNNT2 (NM_001001430.2):c.518G>A (p.Arg173Gln). According to the American College of Medical Genetics (ACMG), the variant was classified as pathogenic/likely pathogenic (class 5-4) which confirmed the clinical suspicion of cardiomyopathy. The NGS variant was also validated by Sanger sequencing. The variant has been previously reported in literature and is associated with the phenotype of cardiomyopathies [7]. Among the 325 genes included in the panel, no other clinically relevant variant of unknown significance, according to ACMG criteria, was identified. A genetic test performed through Next Generation Sequencingdid not find any CNV. However, SNP/CGHarrays were not performed, as the patient withdraw his consent for further investigations. Thus, the relatives were invited to join the cascade program, including both phenotyping (cardiac screening) and genotyping (analysis of the pathogenic variant identified in the proband). Cascade family screening revealed a *TNNT2* variant in the mother (II.1) and his cousin (IV.3). Both of them were phenotype positive (affected by DCM) at cardiac screening at our Hospital clinic. The two sisters of the proband were unavailable for screening. Figure 1 reports both phenotyping and genotyping of the reported family.

## 4. Discussion

To date, the association of dilated cardiomyopathy and repaired COA is very rare. LVSD has been commonly associated with recoarctation in infancy, nonetheless heart failure symptoms seem to regress after hemodynamic relief of aortic gradients [5,6,7,8,9]. Among adult patients, DCM has been commonly reported in association with recoarctation [10,11,12]; similarly, hypokinetic phenotype regressed after stent implantation or surgical repair. In one case, due to severe heart failure, orthotopic heart transplantation [13] was performed. The potential pathophysiologic mechanism of progression to DCM in severe COA in infancy seems to occur after closure of the arterial duct, because formation of an adequate collateral circulation is not allowed promptly [14]. Mechanical wall stress imposed by hemodynamic overload in COA may trigger left ventricular hypertrophy (LVH) and concentric remodeling. Different pathways seem to be involved in adaptive LVH, including increased transcription of sarcomeric proteins, release and production of cytokines and growth factors and production of neurohormonal mediators that contribute to the increase in myocardial mass [15]. Nonetheless, the burden of chronic pressure overload may trigger the compensatory hypertrophy to evolve into maladaptive hypertrophy with cardiac dilatation and loss of systolic function [16,17]. Isthmic recoarctation should be included in the differential diagnosis of children and infants with DCM, regardless of previous surgical or percutaneous correction, and aortic imaging using CMR or CTA is recommended by current Guidelines [18]; nonetheless this was ruled out in our case. Differential diagnosis included myocarditis, acute coronary syndrome or dilated cardiomyopathy (DCM). Absence of hypertensive status and residual isthmic gradient with positive family history of early onset DCM and SCD, suggested a genetic form of cardiomyopathy. Myocarditis was highly excluded on the basis of negative personal history and negative signs on CMR for foci of increased signal compatible with myocardial edema. Genotyping was positive for a heterozygous missense variant in *TNNT2 (NM_001001430.2):* c.518G>A (p.Arg173Gln) which affects a highly conserved domain that encodes for the sarcomeric protein troponin T, and is currently classified as pathogenic/likely pathogenic for hypertrophic cardiomyopathy(HCM), restrictive cardiomyopathy and DCM in several genetic databases [19,20,21,22]. Of note, this variant hasnot been associated to left-sided obstructive lesions according to existing databases. The panel of analyzed genes included genes with causative association with left-sided obstructive lesions, including aortic coarctation. In particular, the proband was tested for NKX2.5, NOTCH1, GATA4, SMAD6 and TBX1 and was found to be negative for pathogenic/likely pathogenic variants.

Cardiomyocytes generated from induced pluripotent stem cells (iPSc) harboring mutations in Troponin T gene may exhibit altered calcium handling and impaired myofilament contraction [23]. Interestingly, iPSc derived cardiomyocytes harboring *TNNT2* mutations seemed more susceptible to β-adrenergic stimulation and morphological analysis showed that their myocardial sarcomeric disarray was improved by treatment with β1 selective-β blocker metoprolol [24].

This case illustrates several issues related to diagnosis and follow up of DCM in the setting of repaired congenital heart defects as COA. Careful history taking, including a family history of cardiomyopathy, should not be overlooked even when the clinical findings seem to suggest a predisposition to hemodynamic factors for LVSD (such as recoarctation). The diagnosis of DCM could not be achieved without detailed family history, including the construction of a three-generation pedigree. Since it is a time-consuming approach, it would not be routinely performed at the time of routine cardiac screening. Referral to an inherited cardiovascular disease center is highly recommended in children and ACHD positive family history for inherited cardiac diseases (cardiomyopathies and genetically determined arrhythmias) since these conditions might be overlooked. Additional genetic testing is essential in the genotyping of family members who may benefit from early diagnosis and regular follow up [25].

## 5. Conclusions

We described a rare case of a young adult with a surgically repaired coarctation of the aorta who complained of mild exertional dyspnea, carrying a heterozygous missense variant in TNNT2 (NM_001001430.2): c.518G>A (p.Arg173Gln). This case highlights the role of careful history taking: a family history of cardiomyopathy should not be overlooked even when the clinical setting seems to suggest a predisposition to hemodynamic factors for LVSD.

## Figures and Tables

**Figure 1 biomolecules-11-00696-f001:**
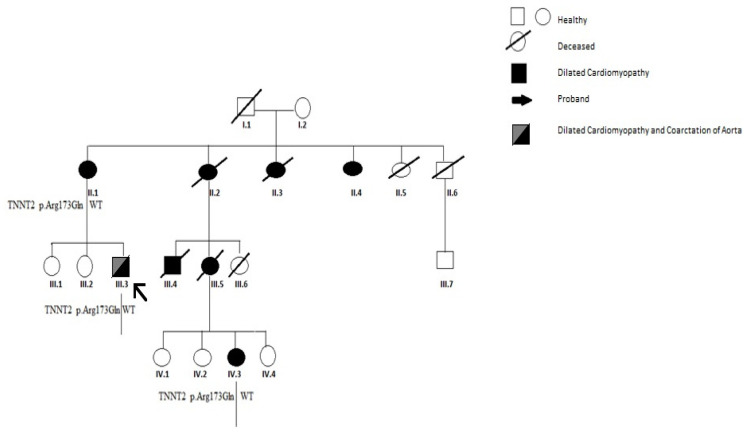
Family pedigree of the proband. Open symbols: phenotype negative. Full symbols: phenotype positive (affected by dilated cardiomyopathy). III,6 died suddenly at 3 months. II,2 III,4 underwent cardiac transplantation. II,2; II,3; III,4; III,5 died aged 42, 18, 24, 38 years old, respectively.

**Figure 2 biomolecules-11-00696-f002:**
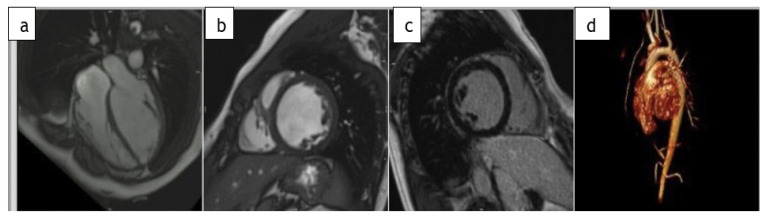
1,5 Tesla cardiac magnetic resonance (CMR) showing a moderately dilated, hypokinetic left ventricle at 4 chamber (**a**) and midventricular short-axis view (**b**); (**c**) shows post contrast short axis view excluding late gadolinium enhancement (LGE). (**d**) Three-dimensional reconstruction of aortic arch where no signs of recoarctation were detected.

## Data Availability

The data that support the findings of this study are available from the corresponding author upon reasonable request.

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
