# Peer review of "Troponin T Mutation as a Cause of Left Ventricular Systolic Dysfunction in a Young Patient with Previous Surgical Correction of Aortic Coarctation"

_biomolecules, 2021, doi:10.3390/biom11050696_

Round 1
Reviewer 1 Report
Auhtors described a case report of DCM and CoA in a family carrying a likely pathogenic rare varinat in TNNT2.
It is a well-described case and only minor points should be addressed:
1.- Any other rare variant in any of other more than 300 genes analyized?
2.- Any CNV identified?
3.- If no history of CoA in any other relative diagnosed with DCM and carrying the TNNT2 variant, what is the potential explanation of CoA? It seems conclusive the association between TNNT2 and DCM in this family but what about CoA?
4.- In panel of analyzed genes analyzed, any gene is currently associated with CoA?
Author Response
1.- Any other rare variant in any of other more than 300 genes analyized?
We thank the reviewer for the useful comment. We confirm that, among the genes included in the cardiomyopathy panel, no other clinically relevant variant of unknown significance, according to ACMG criteria, was identified.
2.- Any CNV identified?
We thank the reviewer for the interesting comment. Genetic test performed with Next Generation Sequencing did not find any CNV. However, we acknowledge that SNP/CGH array were not performed, as the patient withdraw its consent for further investigations. We added these information to the manuscript
3.- If no history of CoA in any other relative diagnosed with DCM and carrying the TNNT2 variant, what is the potential explanation of CoA? It seems conclusive the association between TNNT2 and DCM in this family but what about CoA
We thank the reviewer for the interesting question. We confirm that among family members carrying the TNNT2 mutation, no history of coarctation could be found. Therefore, no causative relationship between missense TNNT2 and coarctation could be demonstrated. The aim of this case report was to underscore the role of a family history of cardiomyopathy in the management of patients with Congenital heart disease. We added a brief comment indicating the absence of a causal relationship in the manuscript
4.- In panel of analyzed genes analyzed, any gene is currently associated with CoA?
We thank the reviewer for the interesting question. The panel of analyzed genes included genes with causative association with left-sided obstructive lesions, including aortic coarctation. In particular, the proband was tested for NKX2.5, NOTCH1, GATA4, SMAD6 and TBX1 and was found to be negative for pathogenic/likely pathogenic variants.
Reviewer 2 Report
The authors present an interesting case of a young male who was diagnosed with left and regular systolic dysfunction secondary to a heterozygous missense variant in TNNT2 many years after surgical correction of coarctation of the aorta. Myocardial dysfunction is seen following coarctation repair but is typically secondary to residual obstruction or recoarctation of the aorta. This case report underscores the importance that LV systolic dysfunction may be from other causes such as a genetic trigger. Overall, this is a nice report which would likely be of interest to the readers of the Journal. However, there are items that should be addressed by the authors prior to any consideration of publication.
- As noted, the manuscript would benefit from careful editing for English usage.
- The authors state that they confidently rule out myocarditis by CMR. However, they did not comment on the extent of CMR reporting to include information about T1 or T2 findings. Please add this information to the manuscript.
- The authors comment on the genetic testing pursued for myocardial dysfunction. Was any testing pursued regarding congenital heart disease or obstructive left-sided heart disease? If so, please add this information. If not, please comment in the manuscript about this limitation.
- The authors have additional information about the initial presentation of the patient regarding coarctation? Was the myocardial function normal prior to and following surgery?
Author Response
REVIEWER 2
Open Review
(x) I would not like to sign my review report
( ) I would like to sign my review report
English language and style
( ) Extensive editing of English language and style required
(x) Moderate English changes required
( ) English language and style are fine/minor spell check required
( ) I don't feel qualified to judge about the English language and style
Yes Can be improved Must be improved Not applicable
Does the introduction provide sufficient background and include all relevant references?
(x) ( ) ( ) ( )
Is the research design appropriate?
(x) ( ) ( ) ( )
Are the methods adequately described?
(x) ( ) ( ) ( )
Are the results clearly presented?
( ) (x) ( ) ( )
Are the conclusions supported by the results?
Nascondi messaggio originale
( ) (x) ( ) ( )
Comments and Suggestions for Authors
The authors present an interesting case of a young male who was diagnosed with left and regular systolic dysfunction secondary to a heterozygous missense variant in TNNT2 many years after surgical correction of coarctation of the aorta. Myocardial dysfunction is seen following coarctation repair but is typically secondary to residual obstruction or recoarctation of the aorta. This case report underscores the importance that LV systolic dysfunction may be from other causes such as a genetic trigger. Overall, this is a nice report which would likely be of interest to the readers of the Journal. However, there are items that should be addressed by the authors prior to any consideration of publication.
As noted, the manuscript would benefit from careful editing for English usage.
The authors state that they confidently rule out myocarditis by CMR. However, they did not comment on the extent of CMR reporting to include information about T1 or T2 findings. Please add this information to the manuscript.
We thank the reviewer for the suggestion. CMR exluded myocarditis and no myocardial edema was detected. We added information about T1 and T2 findings to the manuscript.
The authors comment on the genetic testing pursued for myocardial dysfunction. Was any testing pursued regarding congenital heart disease or obstructive left-sided heart disease? If so, please add this information. If not, please comment in the manuscript about this limitation.
We thank the reviewer for the interesting question. The panel of analyzed genes included genes with causative association with left-sided obstructive lesions, including aortic coarctation. In particular, the proband was tested for NKX2.5, NOTCH1, GATA4, SMAD6 and TBX1 and was found to be negative for pathogenic/likely pathogenic variants
The authors have additional information about the initial presentation of the patient regarding coarctation? Was the myocardial function normal prior to and following surgery?
We thank the reviewer for the question. According to echocardiographic reports, left ventricular ejection fraction was within the reference range prior and following surgery. Therefore, a new onset Left ventricular systolic disfunction seems to be the most likely diagnosis